# Nifedipine Exacerbates Lipogenesis in the Kidney via KIM-1, CD36, and SREBP Upregulation: Implications from an Animal Model for Human Study

**DOI:** 10.3390/ijms21124359

**Published:** 2020-06-19

**Authors:** Yen-Chung Lin, Jhih-Cheng Wang, Mai-Szu Wu, Yuh-Feng Lin, Chang-Rong Chen, Chang-Yu Chen, Kuan-Chou Chen, Chiung-Chi Peng

**Affiliations:** 1Graduate Institute of Clinical Medicine, College of Medicine, Taipei Medical University, Taipei 11031, Taiwan; yclin0229@tmu.edu.tw; 2Division of Nephrology, Department of Internal Medicine, Taipei Medical University Hospital, Taipei 11031, Taiwan; 3Division of Nephrology, Department of Internal Medicine, School of Medicine, College of Medicine, Taipei Medical University, Taipei 11031, Taiwan; maiszuwu@gmail.com (M.-S.W.); linyf@shh.org.tw (Y.-F.L.); 4TMU-Research Center of Urology and Kidney, Taipei Medical University, Taipei 11031, Taiwan; 5Division of Urology, Department of Surgery, Chi-Mei Medical Center, Tainan City 71004, Taiwan; tratadowang@gmail.com; 6Department of Electric Engineering, Southern Taiwan University of Science and Technology, Tainan City 71005, Taiwan; 7Division of Nephrology, Department of Internal Medicine, Taipei Medical University-Shuang Ho Hospital, New Taipei City 23561, Taiwan; 8International Medical Doctor Program, Vita-Salute San Raffaele University, 20132 Milan, Italy; cherylcherylchen@gmail.com; 9Program of Biomedical Sciences, College of Arts and Sciences, California Baptist University, Riverside, CA 92504, USA; eugenechen0529@gmail.com; 10Department of Urology, Taipei Medical University-Shuang Ho Hospital, New Taipei City 23561, Taiwan; 11Department of Urology, School of Medicine, College of Medicine, Taipei Medical University, Taipei 11031, Taiwan

**Keywords:** chronic kidney disease, calcium channel blocker, lipid, SREBP, CD36

## Abstract

Dysregulation of fatty acid oxidation and accumulation of fatty acids can cause kidney injury. Nifedipine modulates lipogenesis-related transcriptional factor SREBP-1/2 in proximal tubular cells by inhibiting the Adenosine 5‘-monophosphate (AMP)-activated protein kinase (AMPK) pathway in vitro. However, the mechanisms by which nifedipine (NF) modulates lipotoxicity in vivo are unclear. Here, we examined the effect of NF in a doxorubicin (DR)-induced kidney injury rat model. Twenty-four Sprague–Dawley rats were divided into control, DR, DR+NF, and high-fat diet (HFD) groups. The DR, DR+NF, and HFD groups showed hypertension and proteinuria. Western blotting and immunohistochemical analysis showed that NF significantly induced TNF-α, CD36, SREBP-1/2, and acetyl-CoA carboxylase expression and renal fibrosis, and reduced fatty acid synthase and AMPK compared to other groups (*p* < 0.05). Additionally, 18 patients with chronic kidney disease (CKD) who received renal transplants were enrolled to examine their graft fibrosis and lipid contents via transient elastography. Low-density lipoprotein levels in patients with CKD strongly correlated with lipid contents and fibrosis in grafted kidneys (*p* < 0.05). Thus, NF may initiate lipogenesis through the SREBP-1/2/AMPK pathway and lipid uptake by CD36 upregulation and aggravate renal fibrosis in vivo. Higher low-density lipoprotein levels may correlate with renal fibrosis and lipid accumulation in grafted kidneys of patients with CKD.

## 1. Introduction

Lipid accumulation in kidney tissue [1,2] causes fatty kidney and lipotoxicity and may result in acute kidney injury, which further transforms into chronic kidney disease (CKD) and renal fibrosis [3]. Although several pathophysiological mechanisms have been proposed, the exact mechanism remains unclear. For example, lipotoxicity is induced by mitochondria dysfunction, endothelium reticulum stress-related insulin resistance, and visceral adiposity [4]. Hyperlipidemia also correlates with calcified atherosclerosis, inducing intimal stenosis and coronary artery disease [5]. Furthermore, reports show that both the impairment of fatty acid oxidation and ATP depletion contribute to renal fibrosis [6]. Increased uptake of fatty acids from the extracellular environment via the cluster of differentiation 36 (CD36) increased activity of transcriptional factor sterol regulatory element-binding proteins (SREBPs), and associated downstream enzymes may also activate lipogenesis [7,8]. Alternatively, reduced activity of AMP-activated protein kinase (AMPK), an energy-sensing molecule that is abundant in mitochondria of the renal tubule and utilizes fatty acids as ATP sources through oxidation, may contribute to lipogenesis [9,10]. Hence, alleviating renal lipotoxicity via regulation of endothelial fatty acid transcytosis, according to a study of a diabetic kidney disease mouse model, may prevent renal dysfunction [11].

Calcium channel blockers, such as nifedipine (NF), can ameliorate obesity-induced revascularization impairment by suppressing oxidative stress [12] and anti-inflammatory activities [13]. The calcium channel blocker benidipine may reduce lipid accumulation in the kidney tubule [14] and intracellular lipid accumulation, which is important in the pathogenesis of kidney fibrosis [15]. However, NF reportedly induces adverse effects on tubular–glomerular feedback, which may cause proteinuria and glomerular hypertension [16,17], as well as lower leg edema [18] and gingival overgrowth [19]. Previously, we showed that NF can upregulate CD36 and fatty acid uptake, trigger reactive oxygen species (ROS) production, and cause de-novo lipogenesis through the AMPK-SREBP transcriptional pathway in renal tubular cells [20]. However, how lipid accumulation affects in-vivo kidney function remains unknown. In this study, we investigated the mechanisms by which lipogenesis and lipotoxicity are modulated by NF in an animal model characterized by proteinuria and hypertension, as well as in patients with CKD who had undergone renal transplantation.

## 2. Results

### 2.1. Changes in Body Weight, Total Cholesterol, Protein Concentration in Urine, and Blood Pressure in the Animal Model

The Sprague–Dawley rats used in this study were divided into four groups: control (fed standard chow), doxorubicin (DR) (DR-induced kidney injury model fed standard chow), DR+NF (DR-induced kidney injury model fed standard chow plus NF), and high-fat diet (HFD) groups (fed a diet containing 60% of calories from fat). From weeks 1 to 7, the body weight (BW) of the HFD group increased rapidly compared to that of the control, DR, and DR+NF groups. The BW of the DR and DR+NF groups increased slowly compared to that in the control group, and the BW change was similar between the DR+NF and DR groups. Doxorubicin comparably suppressed the BW in the DR and DR+N groups (Figure 1A). Urinary protein level was increased in the DR, DR+NF, and HFD groups compared to in the controls at week 7 (*p* < 0.05), with the DR+NF group showing the highest levels (Figure 1B). As shown in Figure 1C, total plasma cholesterol was significantly elevated only in the DR+NF group compared to the DR group (*p* < 0.05). As shown in Figure 1D, the systolic blood pressure of the DR, DR+NF, and HFD groups was significantly higher than that in the control group (*p* < 0.05) and the DR+NF group showed significantly decreased systolic blood pressure compared to the DR group because of NF’s effect. Thus, renal injury was successfully established in the DR and HFD groups, as demonstrated by complications including high proteinuria and blood pressure, observed starting at week 3.

### 2.2. NF Upregulated Tumor Necrosis Factor-α (TNF-α) and Kidney Injury Molecule-1 (KIM-1)

As shown in Figure 2A, compared to the control group, serum TNF-α level was significantly higher in the DR and DR+NF groups (2.46- and 3.08-fold, *p* < 0.05 and < 0.01, respectively) at week 7; in addition, the DR+NF group showed a significantly higher TNF-α level than the DR group (*p* < 0.01). In contrast, the HFD group showed a significantly lower serum TNF-α concentration (0.46-fold, *p* < 0.05) compared to the control. The DR and HFD groups also showed a significant increase in KIM-1 expression in the kidney compared to that in the control group, as indicated by western blot analysis (*p* < 0.05 and *p* < 0.01, respectively) (Figure 2B). Especially, the DR+NF group showed a significantly higher level of KIM-1 than the DR group by both western blot analysis and immunohistochemistry (*p* < 0.1 and *p* < 0.01) (Figure 2B,C). In summary, TNF-α and KIM-1 were significantly elevated following DR-induced kidney injury. Furthermore, the use of NF may exacerbate kidney injury.

### 2.3. Superimposed Damage by NF on Histopathological Lesions of the Kidney

As shown in Figure 3, hematoxylin and eosin (H+E) staining of the renal tissues showed that the DR group had more severe pathological damage compared to the control, as demonstrated by the increased mononuclear cell infiltration, fibrosis, necrosis, and tubular dilatation. The severity was even greater in the DR+NF group (Figure 3A). Although the HFD group also exhibited high levels of mononuclear cell infiltration, necrosis, and dilatation, fibrosis was not apparent in this group. In summary, tubular dilation and mononuclear cell infiltration were observed in all experimental groups but not in the control. However, the DR+NF group exhibited the most dominant interstitial fibrosis, accounting for up to 25% of the lesion degree = 2 (1–25%).

### 2.4. CD36 Expression in Peri-Tubular Membrane upon Immunohistochemical (IHC) Staining

IHC staining of the kidney indicated that peri-tubular membrane CD36 expression was high in the DR groups (Figure 4A); however, expression was further increased in the DR+NF group compared with that in the DR group (*p* < 0.01). Cubilin expression was lower in the DR, DR+NF, and HFD groups than in the control group (*p* < 0.01). Additionally, NF did not cause a further reduction in the cubilin level (Figure 4B). Nile red staining revealed the deposition of more lipid droplets in the DR and DR+NF groups (Figure 4C).

### 2.5. NF Upregulated De-Novo Lipogenesis by Activating SREBP-1/2 Transcriptional Factors and Related Enzymes

The DR+NF group showed higher SREBP-1/2 expression compared to the DR group (*p* < 0.05; Figure 5A). Furthermore, the DR+NF group showed reduced expression of fatty acid synthase (FAS) and acetyl-CoA carboxylase (ACC) compared to the control (*p* < 0.01 and *p* < 0.05, respectively; Figure 5B,C). However, the levels of long-chain fatty acyl elongase (ACSL) and acetyl-CoA synthetase (AceCS1) did not significantly differ among all groups (Figure 5B,C). Additionally, although western blotting showed that the expression of HMG-CoA reductase was seemingly increased in the DR+NF group, and that ATP citrate lyase (ACL) was significantly higher in the HFD group, expressions of these two proteins did not differ between the DR+NF and DR groups (Figure 5D). Immunohistochemistry also revealed the expression of SREBP was mostly localized in the rat renal tubules. The DR and DR+NF groups were shown to increase SREBP. Moreover, nifedipine further aggravated SREBP expression (Figure 5E). In summary, NF treatment significantly induced the expression of SREBP-1/-2, but its related downstream lipogenesis enzymes (ACSL, ACC, AceCS1, ACL and HMG-CoA) were not activated, except for FAS and ACC, which was reduced.

### 2.6. NF Downregulated Phosphorylated AMPK

Phosphorylated ACC (*p*-ACC)/ACC and phosphorylated ACL (p-ACL)/ACL expression was higher in the DR+NF group than in the control group (*p* < 0.05; Figure 6A). Moreover, the DR and DR+NF groups showed similar phosphorylated AMPK (p-AMPK)/AMPK levels (Figure 6B). Finally, levels of the nuclear protein PPARα transcription factor did not significantly differ between the DR+NF and control groups (Figure 6C).

### 2.7. Transient Elastography (TE) on Graft Kidney in Patients with CKD

Transmission elastography analysis revealed that the level of serum low-density cholesterol (LDL) was significantly positively correlated with kidney fibrosis in graft kidneys (r = 0.64, *p* = 0.02; Figure 7A). However, the LDL level only exhibited a trend toward correlating with lipid accumulation (r = 0.06, *p* = 0.05; Figure 7B). Furthermore, serum creatinine levels were positively correlated with kidney fibrosis (r = 0.54, *p* = 0.06; Figure 7C). Therefore, lipogenesis in patients with CKD with graft kidneys is clinically relevant because of its effects on renal function and renal fibrosis.

## 3. Discussion

In this study, we demonstrated a novel effect for NF in augmenting lipogenesis and renal injury in vivo and observed a strong correlation between LDL and tissue fibrosis in patients with CKD who underwent transplant surgery on graft kidneys. A key finding in this study was the observed increase in TNF-α and interstitial fibrosis in rats treated with NF. Further, elevated TNF-α observed in the CKD groups and modulated by NF was not observed in the obesity group, which is consistent with a previous report showing that HFD does not affect the expression of IL-1, IL-6, or TNF-α compared to after feeding of low-fat diets in an animal model [21]. The results of the current study are also consistent with those of our previous in-vitro study [18], in which NF was found to activate SREBP-1/2 and suppress AMPK, leading to lipogenesis and lipid accumulation. An interesting study using HFD mice showed that upregulated AMPK inhibited lipogenic and adipogenic transcriptional factors [22]. Furthermore, a high-energy status, such as that occurring in obesity, and calcium channel blockers, such as NF, may induce a low active form of AMPK (p-AMPK), thereby reducing energy production. However, the phosphorylated form of the lipogenesis enzyme ACC was inhibited in this study, which contrasts with the results of our previous study. Another animal study showed that ACC inhibition in rodents with non-alcoholic fatty liver disease mediates hypertriglycemia [23], and similar effects have been observed in a myocardium animal model, wherein AMPK-dependent ACC phosphorylation did not affect fatty acid oxidation [24]. We also found that NF activated the SREBP-1/2 de-novo lipogenesis transcriptional pathway, similar to in our previous study. In addition, CD36 was upregulated, which may promote the uptake of fatty acids.

Kidney fibrosis, a condition that causes impaired renal function, was increased in the DR+NF group but not in the HFD group. In an earlier literature review, uKIM-1 level correlated with the severity of renal histological damage, and can be a potential reliable predictor of adverse renal outcomes in acute tubular injury [25], and KIM-1 is a sensitive tissue indicator of AKI [26]. TNF-α, which causes inflammatory cell injury, can significantly induce the expression of the apo-A4 protein, which is also associated with pro-inflammatory acute kidney injury in human kidney cells [27]. Similarly, TNF-α was higher in the DR+NF group than in the HFD group. Urine protein and blood pressure (BP) were significantly increased, not only in the DR+NF group but also in the HFD group. The reasons for these results remain unclear. However, renal injury may serve as a more rapid and significant inducer of lipogenesis and adipogenesis. For example, lipid accumulation followed by renal fibrosis was induced in a xanthine oxidoreductase-depleted model [28]. In addition, high-glucose dialysate fluid induced peritoneal fibrosis, which was ameliorated by rapamycin along with a decrease in lipid accumulation in the peritoneum [29]. It is also possible that HFD-fed animals show a delayed renal fibrosis effect. For example, more than two months of HFD treatment has been shown to induce the accumulation of phospholipids in enlarged lysosomes within renal tubular cells [30]. Inflammation of visceral adipose tissue, ectopic fat deposition, and adipose tissue dysfunction may also mediate obesity-related nephropathy independently of total body fat mass [31].

When kidney cells are injured, individual cell types are affected in different ways. Endothelial cells undergo apoptosis, which induces inflammation and mesangial cell proliferation and eventual glomerulopathy. Furthermore, podocyte apoptosis and ER stress may be evoked, resulting in proteinuria and glomerulopathy. Finally, tubular cells also undergo apoptosis with increased autophagy vesicles, leading to interstitial inflammation and fibrosis [32]. How AKI induces renal fibrosis is of great interest in the transition of AKI to CKD [33]. Specific injury to the proximal tubule results in inflammation, and repeated injury leads to tubule injury, chronic inflammation, interstitial myofibroblast proliferation, vascular rarefaction, interstitial fibrosis, and glomerular sclerosis [34]. Intrarenal renin–angiotensin system activation is one of the mechanisms induced in renal fibrosis in FA-induced ER stress in the kidney [35]. In our study, nifedipine may have caused more extensive renal fibrosis in the DR-induced animal model compared to the HFD.

CD36 plays an important role in the uptake of fatty acids into tubular cells, wherein the fatty acids may combine with albumin to induce apoptosis of renal tubular cells [36]. Herein, we report that CD36 was significantly expressed in the NF-treated group, which was consistent with our previous study [18]. Alternatively, similar to the observations for kidney fibrosis, the HFD group showed no CD36 expression. Therefore, HFD-induced renal tubular injury may occur through other mechanisms, such as oxidative stress, activation of the renin–angiotensin system, and dysregulated sodium transporters and circadian clock [37]. An alternate explanation is that HFD suppressed cAMP levels in the liver and in adipose tissue but not in the kidney [38]. However, both the NF-treated and HFD groups exhibited downregulated cubilin levels, which have been previously correlated with inhibition of renal tubular reabsorption [39]. The increased flow of tubular fluid may further activate tubuloglomerular feedback and decrease the glomerular filtration rate [40].

In this study, FAS expression was suppressed, similar to in our previous in-vitro study [17]. FAS generates fatty acids from acetyl-CoA in the cytosol in a process referred to as “de novo fatty acid synthesis,” which uses malonyl-CoA as a substrate. The fatty acid β-oxidation pathway is upregulated by FAS inhibition, and subsequent endothelium dysfunction can be induced by ROS pathways [41]. Based on our findings, NF may improve endothelial function in patients with familial hypercholesterolemia without altering plasma lipids [42] and protect against high-fat diet (HFD)-induced vascular constriction by reducing endothelial nitric oxide synthetase degradation [43].

Although we did not analyze the effect of anti-hypertensive agents in the patients who had undergone a kidney transplant, there is evidence that calcium channel blockers are less beneficial compared to renin–angiotensin–aldosterone system inhibitors [17]. CCBs and RAAS inhibitors have different characteristics in terms of renal protection during anti-hypertension therapy. CCBs inhibit the flow of extracellular calcium through voltage-gated L-type calcium channels, which are responsible for the excitation of smooth and cardiac muscles and aldosterone secretion from the adrenal cortex in humans. When calcium influx is inhibited, vascular smooth muscle cells relax, resulting in vasodilation and BP reduction. However, these two anti-hypertensive agents showed different renal effects, not through their blood-pressure-lowering effects but via rather different effects on lipid accumulation [44] or the vasodilatation of afferent arterioles of the glomerulus [16].

Two major subtypes of CCBs exist and are classified according to different binding sites on the L-type calcium channel: dihydropyridines (e.g., nifedipine and amlodipine) or non-dihydropyridines (e.g., diltiazem and verapamil) [45]. All CCBs are vasodilators that exert blood-pressure-lowering effects across all patient groups, regardless of sex, ethnicity, age, or dietary sodium intake. Nimodipine has been approved for short-term use in patients with subarachnoid hemorrhage but has not been indicated for hypertension treatment. Dihydropyridine CCBs are less likely than non-DHP CCBs to reduce cardiac output because non-DHP CCBs can exert negative inotropic effects. CCBs cause natriuresis by increasing renal blood flow, dilating the afferent arterioles, and increasing glomerular filtration pressure [46]. Dihydropyridine CCBs are highly heterogeneous with respect to their antiproteinuric and renal protection effects, which can be attributed to T receptor blockage in the glomerular efferent arteriole [47]; however, in this regard, nifedipine exerted no benefit.

Patients with CKD had a higher probability of developing hypercholesterolemia-related kidney damage, including decreased expression of vascular endothelial growth factor and graft dysfunction [48], as well as ectopic lipid accumulation in the kidney and its clinical impact except for steatosis of the liver. Interestingly, for cilastatin-induced kidney toxicity in an animal model, lipid imaging was performed by MALDI mass spectrometry of kidney sections. In the study, statin therapy successfully lowered the lipid distribution, which had previously been induced by cisplatin toxicity, and decreased the lipid peroxidation capacity, with statin showing no effect on the lipid distribution in the normal control [49]. This paper described a new link in the pathology between AKI and the lipid distribution. Therefore, methods for detecting lipid accumulation in the kidney tissue are clinically important. Fibroscan is currently the most suitable method for measuring the lipid content of the kidney. We observed lipid accumulation in the kidney and found that it was associated with clinical renal fibrosis and renal function. Thus, nephrologists should consider the lipid profiles of patients with CKD.

There were some limitations to this study. First, we did not include an NF-only animal group to observe its lipogenesis effect. This was because normotensive SD rats may suffer from unexpected acute kidney injury following NF use. Second, the study duration was only 5 weeks, following the induction of kidney injury by doxorubicin. A longer follow-up period should be evaluated to determine the long-term effects of NF on creatinine changes in CKD animals.

In conclusion, lipotoxicity may be a consequence of AKI, and repeated AKI may induce CKD. Additional studies should focus on lipogenesis and lipotoxicity in the kidney and on pioneer treatments for preserving renal function or detecting AKI in advance to prevent CKD.

## 4. Materials and Methods

### 4.1. Experimental Animals

Twenty-four 4-week-old male Sprague–Dawley rats, obtained from the National Laboratory Animal Center, Taipei, Taiwan, were used in this study. The animals, weighing 200–250 g, were housed in plastic cages at 25 ± 1 °C with a 12-h/12-h light/dark cycle and ad-libitum access to food and water. The rats were randomly divided into the following four groups, with six rats per group: control group, rats fed with standard chow; doxorubicin (DR) group, rats in which the DR-induced kidney injury model was established, rats fed with standard chow; DR+NF group, with the establishment of the DR-induced kidney injury model, fed with standard chow plus NF from week 3 onwards; HFD group, rats were fed with a diet containing 60% of its calories in fat. For comparison of the lipidemic status in the kidneys, an HFD was fed to prepare the obese animal model as a positive control in our design, which also presented kidney injury [50] in tubule-interstitial fibrosis [51], lysosomal dysfunction and lipotoxicity [30], glomerulonephropathy, and proximal tubular damage [52]. To establish the DR-induced kidney injury model, a single injection of DR at a dose of 8.5 mg/kg was administered after weighting the rats. Doxorubicin, or Adriamycin, is used for chemotherapy for cancer but shows cardiac toxicity. It specifically damages the podocyte of glomeruli and induces proteinuria and kidney injury, and thus it is a CKD model focused on focal segmental glomerulosclerosis. The usual dose is one injection of 1.5–7.5 mg/kg in rats, and the onset time is approximately 1–2 weeks after injection of doxorubicin, with peak effects observed at 4 weeks [53]. Damage to only podocytes, but also to renal tubular cells or the basement membrane of peri-tubular capillaries, can be found [54]. Four weeks later, all rats were checked for systolic and diastolic BP. The DR+NF group was treated with NF (Adalat OROS^®^, Bayer Taiwan Co., Taipei, Taiwan) corresponding to 0.5 mg/kg/day by referring to the human dose for nifedipine (typically 30–60 mg per day in 60-kg adults).

To measure serum TNF-α levels, blood samples were allowed to clot for 2 h at 36 degrees Celsius before centrifugation at 1000× *g* for 20 min. The serum was collected and assayed immediately, or aliquots were made and stored at ≤−20 °C. The TNF-α concentration was measured using a rat immunoassay (Mouse TNF-alpha Quantikine ELISA kit, MTA00B; R&D Systems, Minneapolis, MN, USA). Kidneys, hearts, adipose tissue and livers were rapidly separated, immediately frozen in liquid nitrogen, and stored at −80 °C for later use. All animal studies were approved by the Institution of Animal Care and Use Committee of Taipei Medical University (LAC-2016-0152, 14 June 2016).

### 4.2. Blood/Urine Sampling and BP Check-Up of Animals

BW at the beginning (W0) of the study and that gained at the end of each week (W1–7) were measured for all animals. Food intake was recorded every 2 days. At W7, blood pressure was measured using the IITC Mouse and Rat Tail Cuff BP System (normal range: 129/91 mmHg) (BioLASCO, Taipei, Taiwan). The serum and urine samples were analyzed using a VetStat^™^ (IDEXX, Westbrook, ME, USA) and VetLab UA (IDEXX). Blood samples were collected once every 3 weeks (W1, W4, and W7) and centrifuged at 1500× *g* for 10 min to obtain the serum. The collected serum was sent for biochemical analyses to determine blood nitrogen urea (normal range: 16 ± 3 mg/dL), creatinine (normal range: 0.4 ± 0.1 mg/dL), total cholesterol (normal range: 74 ± 11 mg/dL), and triglyceride (normal range: 39.42 ± 14.58 mg/dL) levels. Each rat was housed in a metabolic cage with ad-libitum access to food and water for one day to collect urine for determining its protein content after centrifugation at 1000× *g* for 1 min.

### 4.3. Sacrifice of Rats and Histopathological Examination

The rats were anesthetized using 5% isoflurane. The diaphragm was then cut to induce pneumothorax and to permit harvesting of all organs (heart, kidneys, liver, adipose tissue), which were then soaked in 10% formalin fixative for routine histology. Paraffin embedding was performed for further tissue slicing for histology examination. Kidney, heart, adipose tissue, and liver were stored immediately at −80 °C. H & E staining was performed by special technicians from the National Laboratory Animal Center.

All rats received an NC3Rs strategic award for humane techniques used for killing lab rodents. In addition to the experimental work, the dissemination of the results of this research forms an important part of our program and, to that end, is supported by the NC3Rs in the Laboratory Animal Center at Taipei Medical University.

### 4.4. Immunohistochemistry Staining of CD36, KIM-1, and SREBP

The paraffin-embedded sections were deparaffinized before staining. Endogenous peroxidase activity was blocked by 3% hydrogen peroxide for 10 min before treatment with antibodies of anti-CD36 (Taiclone, Taipei, Taiwan), anti-KIM-1 (R&D Systems, Minneapolis, MN, USA) and anti-SREBP (Abcam, Cambridge, UK). Thereafter, horseradish peroxidase recombinant fusion protein anti-rabbit IgG (1:500) was applied. Finally, DAB chromogen (ScyTeck Laboratories, West Logan, UT, USA) was used and images were acquired using a fluorescent microscope (IX81, Olympus, Tokyo, Japan).

### 4.5. Nile Red Staining of Intracellular Lipid Droplets

The paraffin-embedded tissue section was washed with deionized water for 1–2 min and then treated with Nile red staining dye (Sigma-Aldrich, St. Louis, MO, USA), followed by examination under fluorescent microscopy.

### 4.6. Western Blotting

The tissue was harvested from the kidney cortex of the sacrificed animals. Protein was extracted for western blotting from the cortex. The primary antibodies used in this study were Fatty Acid and Lipid Metabolism Antibody Sampler Kit (#8335), anti-TNFR1 (#13377), anti-AMPK (#5832), and anti-p-AMPK (#2535), from Cell Signaling Technology (Danvers, MA, USA); anti-SREBP-1 (sc-13551) and anti-SREBP-2 (sc-13552), from Santa Cruz Biotechnology (Dallas, TX, USA); anti-CD36 (NB400-144ss), anti-β actin (NB600-501), and anti-α tubulin (NB100-690), from Novus Biologicals (Littleton, CO, USA); anti-PPARα (GTX01098), anti-HDAC1 (GTX100513), and anti-histone H3 (GTX122148), from GeneTex (Irvine, CA, USA); and anti-KIM 1 (ab190696, Abcam, Cambridge, UK). The membrane was then subjected to the secondary antibody, either anti-mouse IgG or anti-rabbit IgG (Jackson ImmunoResearch, West Grove, PA, USA), which was dissolved in 5% skim milk in TBST for 1 h. Next, the membrane was incubated for 1–2 min in enhanced chemiluminescence mixture (JT96-K004M, T-Pro Biotechnology, Zhonghe, New Taipei City, Taiwan) for visualization. The western blotting assay was repeated at least three times.

### 4.7. Enrollment of Patients with CKD Who Underwent Kidney Transplantation

The study recruited patients with CKD who underwent kidney transplantation from the out-patient department at Taipei Medical University Hospital. This study involving human subjects was reviewed by the institutional review board committee of Taipei Medical University Hospital. (No. 201707021). The exclusion criteria were age >90 years or patients with discomfort who could not tolerate sonography examinations. Eighteen patients were enrolled and provided signed informed consent. Of the enrolled patients, the average age was 59.4 years, average blood urea nitrogen was 21.5 mg/dL, average creatinine was 1.27 mg/dL, and average daily urine protein was 335 mg per day. Seventy-two percent had hypertension under anti-hypertensive agents, and no patients were currently being administered nifedipine. Their demographic biochemistries were recorded and FibroScan^®^ examination was performed on the kidney. We also performed TE and controlled attenuation parameter (CAP). Biochemical analyses were carried out to determine blood nitrogen urea, creatinine, LDL, and protein levels in the urine.

### 4.8. Transient Elastography (TE)

TE was performed on patients at baseline. The FibroScan^®^ device pulse–echo ultrasound acquisition was used to follow the propagation of the shear wave and to measure its velocity, which is directly related to tissue fibrosis [elastic modulus E expressed as E = 3qV2, where V is the shear velocity and q is the mass density (constant for tissues)]. The detailed procedure has been described previously [55]. TE with CAP is a fast, reliable, repeatable non-invasive method for assessing liver steatosis and fibrosis, showing a high accuracy compared to biopsy [56]. A pioneering study of patients with kidney transplants successfully used the FibroScan to evaluate renal allograft fibrosis [57]. All measurements were conducted by the same technician, blinded to patient data. The patient was asked to lie in a supine position. Renal allograft on traditional ultrasound was performed to define the optimal position for the transducer probe for a preferably orthograde positioning of the transducer on the renal surface and maximal broadness of the parenchyma. The renal allograft ultrasound illustrates the site and volume of measurement, which approximates a cylinder 10 mm wide and 25–30 mm long. The distance between the tip of the transducer and kidney surface was measured. If the distance was <20 mm in sonography, probe ‘M’ was selected (measurement depth of 15–40 mm). If the distance was >20 mm, probe ‘L’ was selected (measurement depth of 20–50 mm). The transducer frequency was 5 MHz. Next, the operator pressed the probe button to begin acquisition. The procedure was repeated until 20 valid measurements were performed in each patient. The success rate was calculated as the number of validated measurements divided by the total number of measurements to achieve at least a 90% success rate. The median value was considered representative of the elastic modulus of the tissue.

### 4.9. Statistical Analysis

Analysis of variance and post-hoc tests using SPSS 14.0 (SPSS, Inc., Chicago, IL, USA) were used to statistically evaluate the data, and results are presented as the mean ± standard deviation. Analysis of variance was chosen to test the difference between four study groups with post-hoc analysis. Two-tailed *p*-values ≤ 0.05 were considered statistically significant. The meaning of the different asterisks is described in the relevant figure legends. We used Spearman’s test to find the associations between the non-pyramid distribution of continuous variables such as LDL, elasticity, CAP, and serum creatinine. Analysis of variance is used to test variation among and between groups for more than 3 variables in a group.

## 5. Conclusions

In this study, we demonstrated that nifedipine may induce renal fibrosis and lipogenesis in a CKD animal model with a higher expression of CD36/SREBP-1/2 signals. Interestingly, hypercholesterolemia, as represented by high LDL levels in patients with kidney transplantation and CKD, was significantly associated with kidney fibrosis and lipid accumulation. Therefore, NF use and lipid content should be closely monitored in patients with CKD.

## Figures and Tables

**Figure 1 ijms-21-04359-f001:**
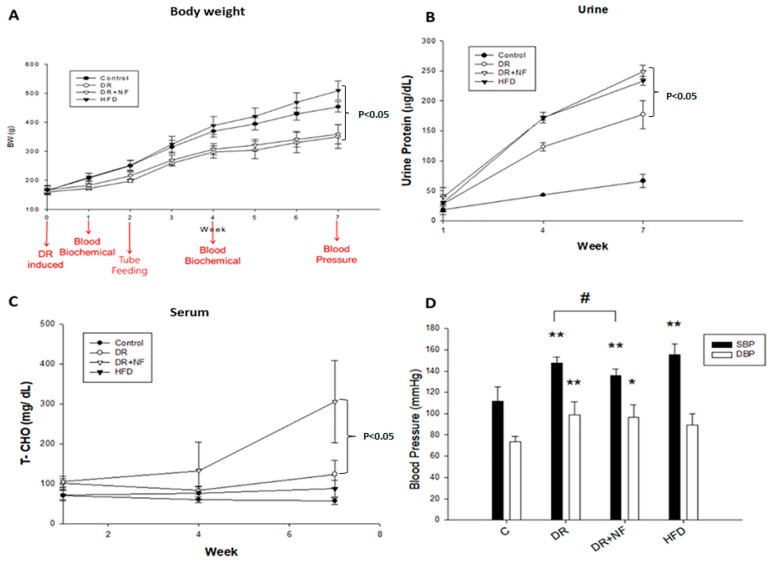
Changes in body weight (BW), urine protein and serum cholesterol levels, and blood pressure (BP) in rats. (*n* = 3 for each group) (**A**) At week 7, the bodyweight of the high-fat diet (HFD) group was higher than that of the DR+NF group (500 vs. 320 g; *p* < 0.05). (**B**) The DR+NF group had a higher urine protein level than the DR group (*p* < 0.05) at week 7. (**C**) The DR+NF group had higher serum total cholesterol than the DR group (*p* < 0.05) at week 7. (**D**) DR+NF, DR, and HFD groups had higher systolic pressure (** *p* < 0.05) compared to the control group. DR+NF and DR groups showed higher diastolic pressure than the control (** *p* < 0.05 and * *p* < 0.1, respectively). The DR+NF group had lower BP compared to the DR group. (^#^
*p* < 0.05).

**Figure 2 ijms-21-04359-f002:**
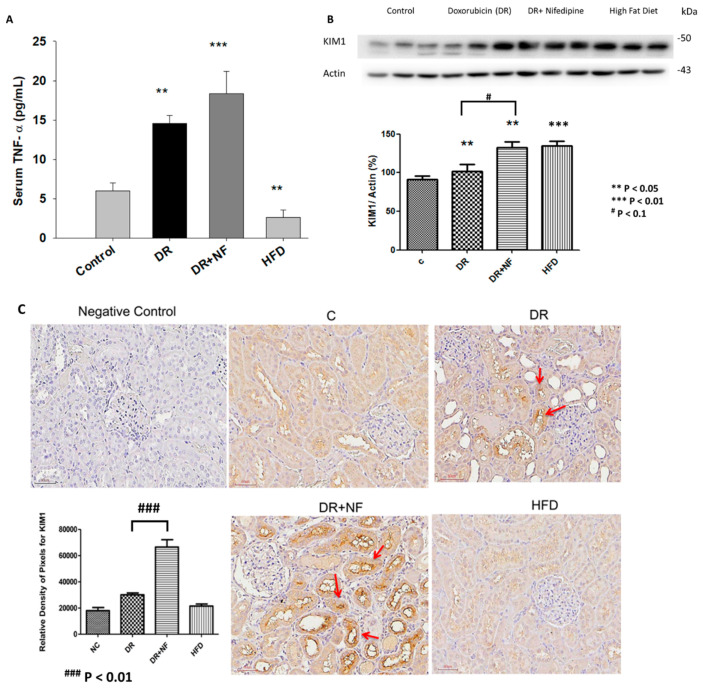
Tumor necrosis factor-α (TNF-α) and kidney injury molecular-1 (KIM-1) shown in the blood sampling and renal tissue of rats. (*n* = 3 for each group) (**A**) ELISA showed that serum TNF-α levels in the DR and DR+NF groups were significantly higher than in the control group (2.46- and 3.08-fold, ** *p* < 0.05 and *** *p* < 0.01, respectively), but the high-fat diet (HFD) group had lower TNF-α levels compared to the control (0.54-fold, ** *p* < 0.05), (**B**) Representative western blotting and quantification of KIM-1 (actin as an internal control ) in renal tissues showed that the KIM-1 of the DR+NF, DR, and HFD group were higher than that in the control group (** *p* < 0.05); besides, the DR+NF group had higher KIM-1 expression than the DR group (1.33-fold, ** *p* < 0.05). (**C**) Immunohistochemical staining and quantification of the image showed a higher intensity of KIM-1 in the renal tubules of the DR and DR+NF group (*** *p* < 0.01). Nifedipine causes much stronger staining of KIM-1 in the DR+NF rats compared to the DR-only group (magnification is 200 ×, ^###^
*p* < 0.01).

**Figure 3 ijms-21-04359-f003:**
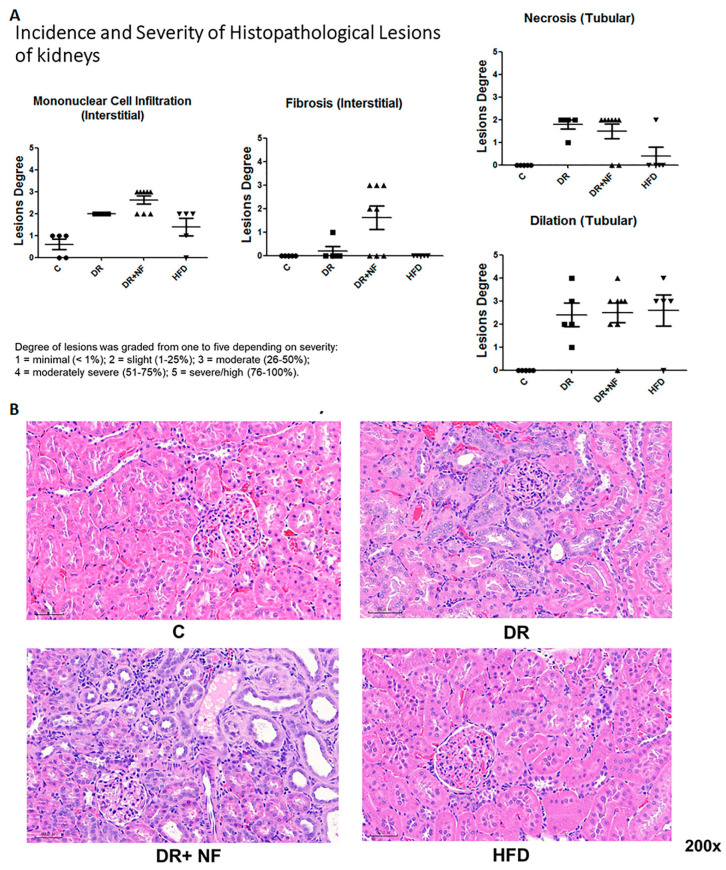
Hematoxylin and eosin (H+E) staining of renal histology in rats (*n* = 3 in each group) (**A**) In renal tissue, DR+NF exhibited slight to moderate lesions (degree = 25–50%), including monocyte cell infiltration, interstitial fibrosis, tubular necrosis, and tubular dilatation compared to the controls. More conspicuously, DR+NF caused more severe fibrosis than DR only (**B**) H+E staining (magnification 200×) showed diffuse cellular infiltration, including lymphocytes and eosinophils (blue) and tubulitis in DR+NF rats, followed by the DR group. Damage observed in the HFD and control groups was minimal.

**Figure 4 ijms-21-04359-f004:**
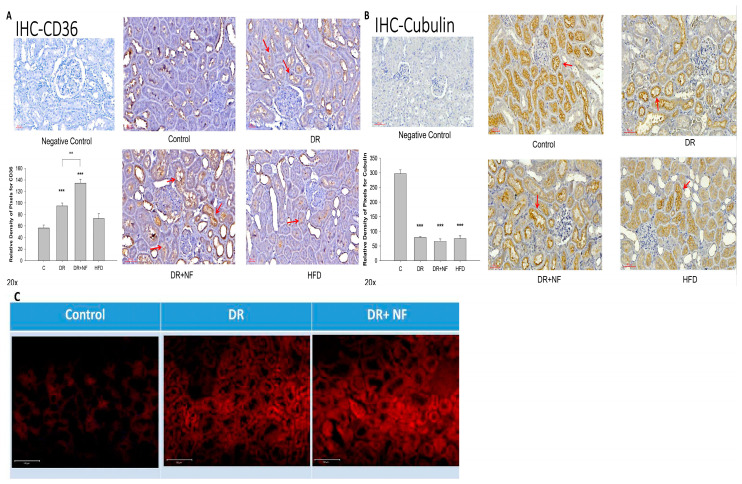
Immunohistochemistry (IHC) of CD36 and cubilin and fluorescent staining for Nile red in rats. (*n* = 3 for each group) (**A**) In kidney tissue, CD36 is highly expressed (red arrow) in the DR group compared to the control. (** *p* < 0.05, *** *p* < 0.01) The DR+NF group also had higher CD36 expression than the DR group. (^##^
*p* < 0.05) (**B**) The expression of cubilin (red arrow) decreased in the DR, DR+NF, and HFD groups compared to that in the control (*** *p* < 0.01). (**C**) Nile red fluorescent staining showed that the DR+NF group had the highest amount of lipid accumulation (red dye), followed by the DR group, compared to the control (magnification 200×).

**Figure 5 ijms-21-04359-f005:**
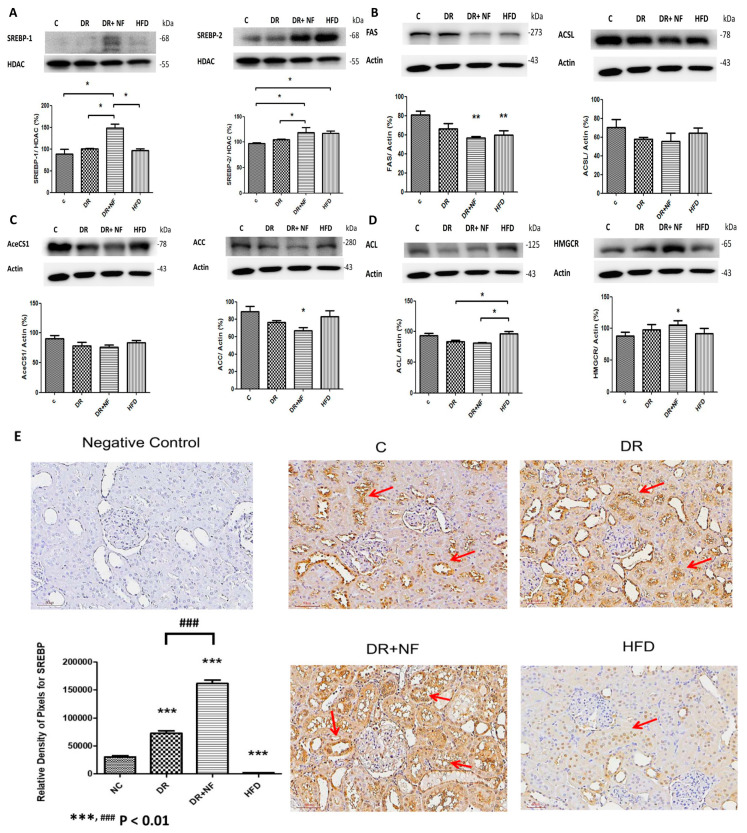
Sterol regulatory element-binding protein (SREBP)-1/2 transcriptional factor and related enzymes in rats. (*n* = 3) (**A**) In renal tissue, western blotting showed that SREBP-1/2 was significantly increased in the DR+NF group compared to the DR group and the control (* *p* < 0.05). (**B**) Fatty acid synthase (FAS) (actin as control) showed lower expression in the DR, DR+NF, and HFD groups compared to the control group (** *p* < 0.01). (**C**) Acetyl-CoA carboxylase (ACC) (actin as control) was decreased in the DR+NF group compared to the control (* *p* < 0.05). (**D**) The HFD group showed considerably higher ACL expression compared to the control group, and the expression of HMGCR was also significant in the DR+NF group relative to the control (* *p* < 0.05). (**E**) The immunohistochemical expression and quantification of SREBP (red arrow) in the rat renal tubule was increased significantly in the DR and DR+NF groups compared to the control. DR+NF was more prominently enhanced than DR only (^###^
*p* < 0.01, *** *p* < 0.01). However, expression of SREBP was decreased in the HFD group (magnification 200×).

**Figure 6 ijms-21-04359-f006:**
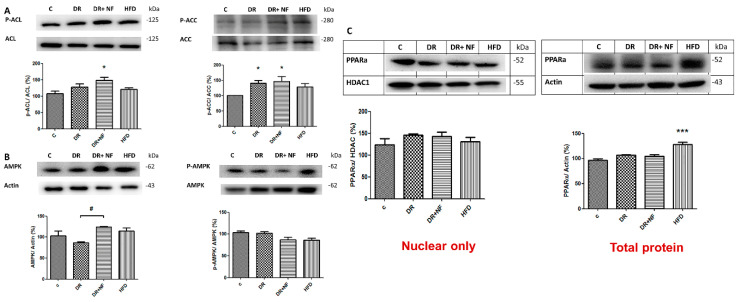
The western blot expression of 5′Adenosine monophosphate-activated protein (AMPK) and peroxisome proliferator-activated receptor 1-α (PPAR-α) in rats. (*n* = 3) (**A**) In renal tissue, phospho-acetyl-CoA carboxylase (P-ACC)/ACC was increased in the DR and DR+NF groups (148% and 150%, respectively, * *p* < 0.05). (**B**) AMPK expression (actin as control) in the DR+NF group was increased relative to the DR group (^#^
*p* < 0.05), whereas phospho-AMPK/AMPK was decreased in DR+NF compared with the control group (^#^
*p* < 0.05). (**C**) Nuclear PPAR-α protein expression levels were similar (HDAC as the internal control). However, total PPAR-α protein was increased by up to 130% in the HFD group compared to the control (actin as control) (*** *p* < 0.05).

**Figure 7 ijms-21-04359-f007:**
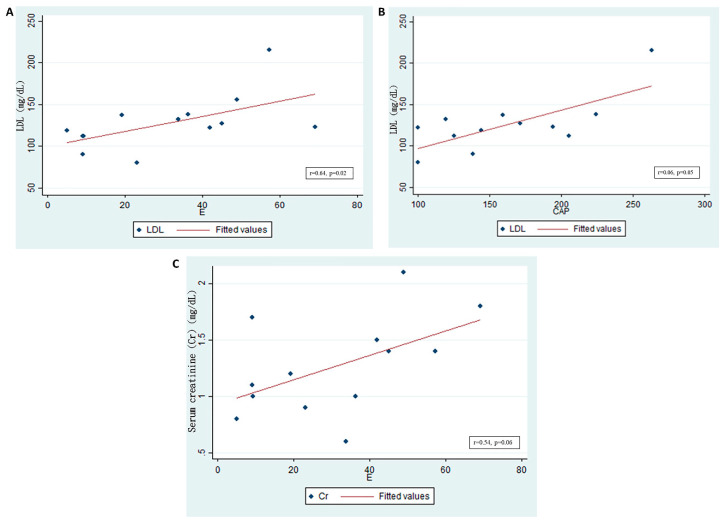
Low-density lipoprotein (LDL) levels and graft kidney lipid accumulation/fibrosis in patients who received kidney transplantation. (*n* = 18) (**A**) Serum LDL significantly, positively correlated with fibrosis in the graft kidneys, as shown by E (kidney stiffness) (r = 0.64, *p* = 0.02). (**B**) Serum LDL significantly, positively correlated with lipid accumulation in the graft kidneys. The controlled attenuation parameter (CAP) (r = 0.06, *p* = 0.05) is shown. (**C**) Serum creatinine (Cr, mg/dL) positively correlated with fibrosis in graft kidneys, as shown by E (r = 0.54, *p* = 0.06).

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
