# Peer review of "Nifedipine Exacerbates Lipogenesis in the Kidney via KIM-1, CD36, and SREBP Upregulation: Implications from an Animal Model for Human Study"

_ijms, 2020, doi:10.3390/ijms21124359_

Round 1

Reviewer 1 Report

The manuscript entitled “Nifedipine Modulates Lipogenesis and Lipotoxicity in Kidney via the AMPK-SREBP Transcriptional Pathway: From Animal Model to Human Study” aims to demonstrate that the calcium channel blocker nifedipine modulates lipogenesis and lipotoxicity by inducing de novo lipogenesis and reducing AMPK signaling. The authors also correlated the level of LDL with the degree of kidney fibrosis in patients with chronic kidney disease. To follow my comments and concerns:

  1. The aim of the study is not clearly stated; thus, it is difficult to understand the conclusions of the authors.
  2. Please explain the rationale for the experimental design. Why an HFD model has been included? Explain it in the text.
  3. The conclusions are based on very small differences obtained from western blotting quantification, which per se are not accurate. I would not feel comfortable to draw any conclusion based on such small differences. I suggest the authors to confirm the key data with other methods. For instance, an ELISA to measure Kim-1 should be feasible as the urine has been collected in metabolic cages. Perhaps some real time RT-PCR might help to support the protein data.
  4. I suggest the authors to rewrite the figure legend. As it is, it is more a repetition of the result section. The figure legend should simply provide to the reader the information required to understand the figure. The authors should avoid data interpretation.
  5. In each figure legend please clarify the statistical test employed. As it is, it is unclear whether the authors compared groups with Student t-test or they used ANOVA followed by a post-hoc analysis (recommended).
  6. The quality of figure 3 is poor so that the figure is difficult to interpret. Please provide a better, perhaps enlarged figure.
  7. I suggest shortening the discussion and make it more incisive. The authors provide a lot of information that can distract the readers from the core of the project.
  8. Please rewrite the conclusions. The sentence “In this study, we demonstrated that NF may increase… but not in an HFD-induced obesity model”. It seems you treated HFD animals with NF, which is not true.

Reviewer 2 Report

 Authors adequately answered to every question and described the limitation of this study.

Round 2

Reviewer 1 Report

I  acknowledge  that the authors followed some of my suggestion. However,

1) I am still convinced that some additional experiments should be done. WB alone are not sufficient.

2)The authors keep avoiding changes in the legend. They change the title but also the content should follow the similar style. Moreover some info are missing , such as the n for each experiment.

Author Response

2020, June 16

Prof. Dr. Maurizio Battino

Editor-in-Chief

Dear Editor:

We wish to re-submit the manuscript titled “Nifedipine Exacerbates Lipogenesis in the Kidney via KIM-1, CD36, and SREBP Upregulation: Implications from Animal Model to Human Study” The manuscript ID is IJMS-791237.

We thank you and the reviewers for your thoughtful suggestions and insights. The manuscript has benefited from these insightful suggestions. Additional experiments were performed following the reviewers’ valuable advice. I look forward to working with you and the reviewers to move this manuscript closer to publication in the International Journal of Molecular Sciences.

The manuscript has been rechecked and the necessary changes have been made in accordance with the reviewers’ suggestions. Note that in the revised version, all changes are highlighted in yellow. The responses to all comments have been prepared and given below.

Thank you for your consideration. I look forward to hearing from you.

Sincerely yours,

Professor Chiung-Chi Peng, PhD

Comments and Suggestions for Authors

1) I am still convinced that some additional experiments should be done. WB alone are not sufficient.

Ans:Thanks for your great suggestions. We have done series of experiments including immunohistochemistry of KIM-1 in figure 2c and immunohistochemistry of SREBP in figure 5E. In addition, we have changed our title to fit accurately with our finding and scopes. “Nifedipine Exacerbates Lipogenesis in the Kidney via KIM-1, CD36, and SREBP Upregulation: Implications from Animal Model to Human Study” to fit our scope in this manuscript.

Therefore, we revised page 4, line 121-125; page 5, line 136-139; page 7, line 183-186 and page 9, line 202-205.

2)The authors keep avoiding changes in the legend. They change the title but also the content should follow the similar style. Moreover some info are missing , such as the n for each experiment.

Ans: Thanks for your great comments. We have carefully revised our figure legends by adding n for each experiments, and figure legends.

Therefore, we revised figure 1 legends from page 3, line 109-115; Figure 2 from page 4, line 129-136; Figure 3 from page 6, line 152-158; Figure 4 from page 7, line 169-175 and line 186-188; Figure 5 from page 8, line 194-205; Figure 6 from page 9, line 215-222; page 10, line 232-237.